# Does tinnitus and emotional distress influence central auditory processing? A comparison of acute and chronic tinnitus in normal-hearing individuals

Qian Zhou[1,2,3,4☯], Wenling Jiang[1,2,3,4☯], Haibin Sheng[1,2,3,4☯], Qinjie Zhang[1,2,3], Dian Jin[1], Haifeng Li[1,2,3], Meiping Huang[1,2,3,4], Lu Yang[1,2,3], Yan Ren[1,2,3], Zhiwu Huang◉[1,2,3,4*]

1 Department of Otolaryngology-Head and Neck Surgery, Shanghai Ninth People's Hospital, Shanghai Jiao Tong University School of Medicine, Shanghai, China, 2 Faculty of Hearing and Speech Science, College of Health Science and Technology, Shanghai Jiao Tong University School of Medicine, Shanghai, China, 3 Shanghai Key Laboratory of Translational Medicine on Ear and Nose diseases, Shanghai, China, 4 Ear Institute, Shanghai Jiao Tong University School of Medicine, Shanghai, China

☯ These authors contributed equally to this work.
* huangzw86@126.com

## Abstract

### Objective

Tinnitus is characterized by the perception of sound without an external source and can cause auditory difficulties even in individuals with normal hearing.

### Design

This study compared the auditory afferent and efferent functions of normal-hearing patients with acute tinnitus, chronic tinnitus, and control group using gap detection test, the Mandarin Hearing in Noise Test (MHINT), and contralateral suppression of transient evoked otoacoustic emissions (TEOAEs).

### Results

It reveals the neural and emotional dynamics as tinnitus progresses from acute to chronic stages. Patients with acute and chronic tinnitus exhibited reduced contralateral suppression of TEOAEs, elevated gap detection thresholds, and higher speech recognition thresholds in noisy environments, indicating that tinnitus interferes with both afferent and efferent auditory pathways. The lack of significant differences between acute tinnitus and chronic tinnitus patients suggests that auditory functions do not necessarily deteriorate over time, suggesting that tinnitus may not worsen due to neuroplasticity as the condition progresses. The speech recognition ability in noise of patients with acute tinnitus is influenced by emotional scores, implying that emotional distress plays a crucial role in amplifying tinnitus-related interference. When

**Data availability statement:** All relevant data are within the manuscript and its Supporting Information files.

**Funding:** This work was supported by the Shanghai Science and Technology Commission [Grant number 22Y11902000]; the Shanghai Jiao Tong University Medical College Affiliated Ninth People's Hospital Horizontal Project [Grant number JY2023-004]. All funds were received by Dr. Zhiwu Huang.

**Competing interests:** NO authors have competing interests.

emotional burden is reduced, the negative feedback loop between emotional distress and neuroplasticity can be broken, preventing further decline in central auditory function.

## Conclusions

These findings highlight the importance of psychological support and emotional management in clinical practice to improve auditory performance and potentially halt the intensification of tinnitus.

## 1. Introduction

Tinnitus is the perception of sound without any external source, affecting approximately 10% to 25% of adults [1,2]. Beyond being a common auditory symptom, tinnitus can lead to severe issues such as sleep disturbances, anxiety, depression, and cognitive decline [3]. The chronic nature of tinnitus significantly impacts the quality of life for many sufferers, making it an important public health issue. For some patients, the intrusive nature of tinnitus leads to long-term emotional distress and cognitive burdens, especially in environments with background noise, where auditory processing becomes particularly challenging.

The underlying mechanisms of tinnitus are complex, involving both central and peripheral auditory pathways. Recent studies suggest that tinnitus is not only linked to auditory system abnormalities but also to impairments in cognitive processing, working memory, and selective attention during cross-modal integration [4–8]. These cognitive deficits indicate a broader dysfunction within the central nervous system, particularly affecting selective attention. Some studies have found reduced contralateral suppression of otoacoustic emissions in chronic tinnitus patients, suggesting damage to efferent auditory pathways [9,10]. Additionally, other research has shown that tinnitus patients exhibit poorer speech recognition in noisy environments, even without measurable hearing loss [11,12]. However, findings from gap detection tests (GDT) indicate no significant differences in temporal resolution between tinnitus patients and the general population [13], suggesting that afferent pathway damage does not always correlate with impaired temporal processing. It has also been noted that, after controlling for factors like age and hearing loss, tinnitus does not typically affect the perception of external sounds [14]. This suggests that tinnitus (top-down processing) and external sounds (bottom-up processing) might follow distinct neural pathways. Despite the unclear relationship between tinnitus and central auditory processing disorders, tests such as contralateral suppression of otoacoustic emissions, GDT, and speech-in-noise recognition are effective tools for assessing afferent and efferent auditory functions in tinnitus patients. These tests provide a comprehensive understanding of how tinnitus affects both central and peripheral auditory pathways.

Neuroplasticity refers to the structural or functional changes in the brain or nervous system in response to injury or stimulation. A leading model of tinnitus attributes the

phantom percept to maladaptive plasticity within central auditory pathways that compensate for reduced peripheral input by "filling in" missing sound, thereby generating a persistent internal signal [15]. When peripheral input is diminished, these central circuits up-regulate neural gain or reorganize frequency maps, creating a self-sustaining tinnitus percept. This maladaptive plasticity does not act in isolation: auditory regions is tightly interconnected with limbic structures such as the amygdala, hippocampus and anterior cingulate cortex; neuro-imaging shows that tinnitus distress is accompanied by enhanced amygdala-to-auditory-cortex connectivity and broader limbic-auditory coupling [16,17]. Emotion can interfere with cognitive functions such as attention and working memory [18], thereby affecting central auditory processing. These findings provide a biological basis for emotional modulation of auditory plasticity: stress and anxiety heighten attention to tinnitus, repeatedly activating and reinforcing the maladaptive loop, whereas effective emotion regulation may allow the system to recalibrate. Accordingly, we hypothesize that emotional distress can accelerate tinnitus-related neuroplastic changes, potentially creating a vicious cycle, while alleviating the emotional burden might slow or even reverse this process.

Few studies have explored the complex relationship between emotions, neuroplasticity, and tinnitus [16,19]. Most clinical studies have focused on chronic tinnitus [11,14], while few have examined acute tinnitus [10,17,20,21]. Clinically, tinnitus persisting for < 3 months is considered acute, while tinnitus lasting ≥ 3 months is classified as chronic [22]. Acute tinnitus is often linked to recent triggers such as noise trauma, ototoxic medication, or acute stress and may remit if the trigger resolves. By contrast, chronic tinnitus usually reflects more enduring cochlear damage and long-term neurophysiological change [21]. Comparing acute and chronic tinnitus can reveal critical differences and provide insights into how tinnitus evolves. Acute tinnitus patients typically exhibit fewer signs of outer hair cell damage, as indicated by distortion product otoacoustic emissions (DPOAE) recordings, compared to chronic tinnitus patients [10]. In contrast, chronic tinnitus is associated with sustained neuroplastic reorganization—including stronger limbic–auditory connectivity (parahippocampal gyrus, amygdala), cortical hyperactivity, and altered attention networks [21,23,24]. Clinically, acute tinnitus is often accompanied by anxiety, heightened tinnitus awareness, and poor sleep, while chronic tinnitus is more frequently linked to depressive symptoms [20]. However, the changes in neuroplasticity during the transition from acute to chronic tinnitus, as well as the role emotions play in this process, remain unknown.

To our knowledge, this is the first study to investigate differences in central auditory processing function between patients with acute (<3 months) and chronic (>3 months) tinnitus. We employed three behavioral tests—Gap Detection Test (GDT) for temporal resolution, the Mandarin Hearing-in-Noise Test (MHINT) for speech-in-noise recognition, and contralateral suppression of transient evoked otoacoustic emissions (TEOAEs) for efferent-pathway function—to normal-hearing participants. Emotional distress was quantified with the Tinnitus Handicap Inventory (THI) and a tinnitus loudness visual-analogue scale (VAS). By correlating THI/VAS scores with auditory outcomes, we infer how distress may modulate function; differences between acute and chronic groups are interpreted as behavioral markers of underlying neuroplastic adaptation. The purpose of this study is to (i) determine whether tinnitus per se alters central auditory processing, (ii) assess whether these alterations differ between acute and chronic stages, and (iii) examine the relationship between emotional distress and auditory performance. Understanding these links will provide valuable insights into how tinnitus evolves and offer important clinical implications for managing tinnitus.

## 2. Materials and methods

### 2.1 Participants

The study, conducted from May 1, 2023, to May 1, 2024, involved 30 individuals with acute subjective tinnitus (duration <3 months; mean duration ≈ 0.6 months, range 3 days–2.5 months), 30 with chronic subjective tinnitus (duration >3 months; mean duration ≈ 2.6 years, range 4 months–10 years), and 30 healthy volunteers as controls (Table 1). We selected a 3-month cutoff based on clinical guideline to distinguish acute (early-stage) tinnitus from chronic tinnitus [22]. All participants had normal hearing as defined below. Participants with tinnitus were recruited from our hospital's otolaryngology

**Table 1. Characteristics of the study population.**

| | Acute tinnitus group | Chronic tinnitus group | Control group | p value |
|---|---|---|---|---|
| Number (pts) | 30 | 30 | 30 | |
| Age±SD (years) | 31.70+7.71 | 30.07+8.09 | 28.07+8.15 | 0.216 |
| PTA (dB HL) | 9.71+4.39 | 8.54+3.61 | 8.08+4.49 | 0.251 |
| Tinnitus left (pts) | 7 | 6 | — | |
| Tinnitus right (pts) | 13 | 3 | — | |
| Tinnitus bilateral (pts) | 10 | 21 | — | |
| Non-tinnitus ears | 20 | 9 | 60 | |
| THI (scores) | 43.33+18.86 | 31.87+18.84 | | 0.038 |
| Functional | 16.57+9.08 | 12.40+8.97 | | 0.112 |
| Emotional | 14.00+7.51 | 9.67+7.37 | | 0.047 |
| Catastrophic | 12.76+4.35 | 10.40+4.53 | | 0.080 |
| VAS (scores) | 5.00+1.67 | 4.47+1.80 | | 0.288 |

Note. Age and PTA p value from one-way ANOVA (three groups); THI and VAS p values from independent-samples t test (acute vs chronic tinnitus only).

clinic, and each had their tinnitus diagnosis confirmed via clinical assessment (including history and tinnitus matching). Chronic tinnitus patients in our sample had experienced tinnitus for a mean duration of approximately 2.6 years (ranging from ~4 months to 10 years). Control volunteers were age-matched individuals with normal hearing and no history or complaint of tinnitus. They were recruited by advertisement among hospital staff, relatives, and community members, and interviewed to ensure they had no perceived tinnitus; therefore, they did not complete THI or VAS questionnaires. All participants were aged between 18 and 55 years, as we restricted the age range to avoid the effects of presbycusis on the results.

Participants underwent otoscopic examination, tympanometry (0.226 kHz), and standard pure-tone audiometry to rule out middle ear or retrocochlear pathologies. All had normal tympanograms (Type A) and underwent pitch matching to determine tinnitus frequency and loudness matching for tinnitus loudness assessment. Participants with tinnitus completed the Tinnitus Handicap Inventory (THI) [25], and provided a self-assessment of tinnitus loudness using a Visual Analogue Scale (VAS), ranging from 0 to 10, with higher scores reflecting increased perceived loudness. Pure-tone thresholds were measured at 0.25–8 kHz with a calibrated audiometer (Madsen Astera) and insert earphones, following ISO 7029 standards. Normal hearing was defined as ≤ 20 dB HL at all tested frequencies; tympanometry confirmed type-A tympanograms in every ear. All participants had normal hearing, were of similar age, and had no history of excessive noise exposure, ototoxic medication, chronic middle ear disease, or familial hearing loss. Not every participant was able to complete every behavioral task because of time constraints or fatigue; therefore, the effective sample size (n) varies by test. The exact n for each analysis is provided in the corresponding figure or table legend.

This study was approved by the Ethics Committee of the Shanghai Ninth People's Hospital (Approval number: SH9H-2–22-T379-1), located in Shanghai, China. All participants were adults aged 18 years or older, and written informed consent was obtained from each participant prior to inclusion in the study.

## 2.2 Tinnitus severity evaluation

Tinnitus severity was measured using the Tinnitus Handicap Inventory (THI) and Visual Analogue Scale (VAS). The THI is a widely used 25-item questionnaire that assesses the impact of tinnitus on daily life on three dimensions: functional, emotional and catastrophic. It is a three-item choice, with yes representing a 4, sometimes a 2, and never a 0. The total score ranges from 0 to 100; higher scores indicate more severe tinnitus [26]. The VAS was used to quantify subjective tinnitus loudness, with participants rating their tinnitus from 0 (no tinnitus) to 10 (extremely loud tinnitus).

## 2.3 Auditory function tests

Auditory function tests included air conduction pure-tone audiometry, distortion product otoacoustic emissions (DPOAE), contralateral suppression (CS) of transient evoked otoacoustic emissions (TEOAEs), gap detection test (GDT), and Mandarin Hearing in Noise Test (MHINT). Certified audiologists conducted these evaluations in a soundproof and electromagnetically shielded laboratory, with background noise levels below 25 dB (A).

## 2.4 Pure-tone audiometry

Pure-tone audiometry was conducted at frequencies of 0.25, 0.5, 1, 2, 3, 4, 6, and 8 kHz using an audiometer fitted with insert earphones (Madsen Astera, GN Otometrics, Denmark), following the ISO 8253–1:2010 guidelines. All participants demonstrated pure-tone hearing thresholds of 20 dB HL or better across the standard audiometric frequencies.

## 2.5 Distortion Product Otoacoustic Emissions (DPOAE)

DPOAE testing was performed using a cochlear emission analyzer (Capella, GN Otometrics, Denmark). The results were deemed valid if the emission amplitude surpassed the noise floor by at least 3 dB. DPOAE were elicited using two primary tones (L1 = 65 dB SPL, L2 = 55 dB SPL) with an f2/f1 ratio of 1.22. The 2f1–f2 distortion product (DP1) was recorded for each stimulus pair at frequencies of 1, 2, 3, 4, 6, and 8 kHz. Before each recording, the probe placement and tone levels were carefully checked and confirmed.

## 2.6 Contralateral suppression of TEOAEs

Contralateral suppression (CS) of TEOAEs was used to assess the medial olivocochlear (MOC) efferent reflex [27]. TEOAEs were elicited using linear click stimuli at 60 dB peak equivalent SPL (peSPL) at a rate of 19.3 clicks per second, both with and without a 50 dB SL contralateral white noise suppressor (i.e., 50 dB above each participant's white noise threshold) delivered through an audiometer and insert earphones, while keeping the probe in place. The suppressor intensity was set below the stapedius reflex threshold for all participants. Responses were averaged over 2,080 sweeps, ensuring at least 90% stimulus stability. Suppression was calculated by subtracting the TEOAE amplitude during contralateral noise stimulation from the amplitude without stimulation, at frequencies of 1, 2, 3, 4, and 5 kHz.

## 2.7 Mandarin Hearing in Noise Test (MHINT)

The MHINT [28] comprises 12 lists, each with 20 sentences, and each sentence contains 10 Chinese characters. Sentences were presented at various signal-to-noise ratios (SNRs) using BLIMP software (version 1.3, House Ear Institute, USA) via a personal computer and headphones (HD200, Sennheiser, Germany). Testing followed an adaptive procedure [29], starting with 60 dB (A) fixed ipsilateral white noise, and the first sentence presented at 5 dB SNR. Participants repeated sentences accurately, and SNR was calculated as the presentation level at which the subject correctly identified 50% of the sentences. The speech recognition threshold (SRT) was obtained as the final result, with lower SRT values indicating better speech recognition ability in noise.

## 2.8 Gap Detection Test (GDT)

The Gap Detection Test (GDT), measuring temporal resolution in milliseconds (ms), was assessed using a three-interval forced-choice procedure. We used a custom MATLAB (v7.0) program to present the stimuli via ER-3A insert earphones in a soundproof booth, at 40 dB SL (i.e., 40 dB above each participant's threshold for the stimulus). Each trial included three 1000 ms narrow-band Gaussian noise bursts (either centered at 1 kHz or 4 kHz; for tinnitus patients, one test used a noise band centered at their individual tinnitus frequency in lieu of 4 kHz). One of the three bursts contained a brief silent gap. The silent gap had an abrupt onset and offset (no ramping applied). The initial gap duration was 20 ms, and it decreased

in 2-ms steps between trials until the participant failed to detect the gap. A two-down/one-up adaptive rule (1-ms step size) then refined the threshold. The participant's task was to identify which burst had the gap, and the gap detection threshold was defined as the shortest interval where participants consistently detected the gap [30].

### 2.9 Statistical analysis

Descriptive statistics were used to summarize participant characteristics. Normality was assessed using Kolmogorov–Smirnov tests. For normally distributed data, group differences in THI, VAS scores, age, DPOAE amplitudes, CS on TEOAEs, MHINT SRT, and gap detection thresholds were analyzed using independent sample t-tests and ANOVA. For non-normal data, Mann-Whitney U and Kruskal-Wallis H tests were employed. Chi-square tests were used to examine group differences for ordinal data. Spearman correlation coefficients analyzed the relationships between auditory function and tinnitus characteristics. All statistical analyses were performed using SPSS version 26.0, with $p$-values $<0.05$ considered significant.

## 3 Results

### 3.1 Characteristics of subjects

Table 1 provides a summary of tinnitus characteristics for both the acute and chronic tinnitus group, along with the audiological findings in each group. For patients with unilateral tinnitus, the dominant ear refers to the ear experiencing tinnitus, while for those with bilateral tinnitus, the dominant ear is the one with louder tinnitus. The analyses for TEOAEs and MHINT tests included the effect of the dominant ear on tinnitus patients. No significant differences in age were found among the three groups (one-way ANOVA, $F(2,87) = 1.558$, $p=0.216$). All participants met the $\leq 20$ dB HL criterion for normal hearing (see Table 1). Mean PTA did not differ significantly among acute tinnitus, chronic tinnitus, and control groups ($F(2, 87) = 1.405$, $p=0.251$). The acute-tinnitus group (n=21) showed a significantly higher total THI score than the chronic-tinnitus group (n=30) ($t(49) = 2.138$, $p=0.038$), driven largely by a difference in the Emotional subscale ($t(49) = 2.043$, $p=0.047$). No significant differences were observed in the functional and catastrophic domains ($t(49) = 1.623$, $p=0.112$; $t(49) = 1.787$, $p=0.08$, respectively). There was no significant difference between the acute and chronic tinnitus groups in the VAS scores ($t(49) = 1.073$, $p=0.288$)。

### 3.2 Comparison of auditory functions between acute tinnitus, chronic tinnitus, and control group

We conducted separate one-way ANOVAs at each of the six test frequencies (1–8 kHz) and found no significant group differences in DPOAE DP1 amplitudes at any frequency between the three groups (all $F(2,75) \leq 1.684$, $p\geq 0.193$; see Fig 1A). Group differences among three groups in contralateral suppression (CS) of TEOAEs are shown in Fig 1B. Significant differences were found in the CS on TEOAEs at 1 kHz, 2 kHz, and in the average values between the three groups ($F(2,67) = 6.426$, $p=0.003$; $F(2,67) = 7.467$, $p=0.001$; $F(2,67) = 10.896$, $p<0.001$). Post hoc t-tests with Bonferroni correction showed that both acute and chronic tinnitus groups had significantly lower CS compared to controls ($p<0.05$), but no significant difference between the acute group and chronic group ($p>0.05$).

See Table 2 for the ear-specific values within the tinnitus groups. At each test frequency (1–5kHz and average), we ran a two-way ANOVA with factors group (acute vs. chronic) and ear (dominant vs. non-dominant). No significant main effects of group (all $F(1,28)\leq 0.637$, $p\geq 0.432$) or ear (all $F(1,28)\leq 1.496$, $p\geq 0.232$) were observed, and the group × ear interaction was likewise non-significant at every frequency (all $F(1,28) \leq 2.601$, $p\geq 0.118$).

Fig 2 illustrates the gap detection thresholds for the three groups. A two-way ANOVA (group × frequency) revealed significant effects of both group ($F(2,156) = 11.155$, $p<0.001$) and frequency ($F(1,156) = 280.765$, $p<0.001$), along with a significant interaction between group and frequency ($F(2,156) = 3.300$, $p=0.039$). Post hoc t-tests with Bonferroni correction revealed that both tinnitus groups had significantly higher gap detection thresholds compared to controls ($p<0.01$), but there was no significant difference between the acute tinnitus and chronic tinnitus groups ($p>0.05$).

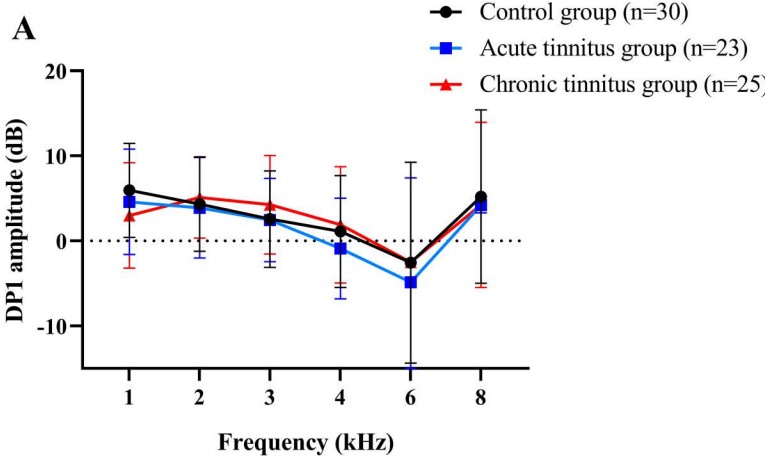

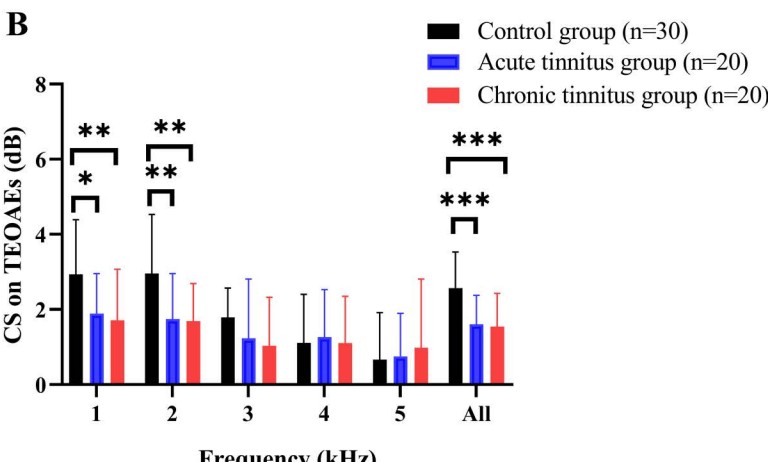

**Fig 1. Effects of groups on DPOAE amplitudes and CS on TEOAEs.** There were no statistically significant differences in the DPOAE amplitudes (average left ear and right ear) at frequencies of 1–8 kHz between three groups (A). There were statistically significant differences in CS on TEOAEs (average left ear and right ear) at 1 kHz, 2 kHz frequencies and average value between three groups (B). Analyses were performed by the one-way ANOVA. *$p < 0.05$, **$p < 0.01$, ***$p < 0.001$.

**Table 2. Contralateral suppression (CS) on TEOAEs for dominant and non-dominant ears in acute and chronic tinnitus groups.**

| CS on TEOAEs (dB) | Acute tinnitus group (n = 14) | | Chronic tinnitus group (n = 16) | |
|---|---|---|---|---|
| | Dominant ears | Non-dominant ears | Dominant ears | Non-dominant ears |
| 1 kHz | 1.62±0.99 | 1.96±1.45 | 1.82±1.33 | 1.80±1.76 |
| 2 kHz | 1.57±1.64 | 1.68±1.73 | 1.79±0.93 | 1.60±1.15 |
| 3 kHz | 0.84±2.77 | 1.71±1.38 | 1.26±1.58 | 1.21±1.21 |
| 4 kHz | 1.17±2.06 | 1.41±1.35 | 1.56±2.09 | 0.82±0.86 |
| 5 kHz | 0.74±1.80 | 0.56±1.37 | 0.88±2.69 | 1.40±2.45 |
| All | 1.53±0.95 | 1.91±1.09 | 1.70±0.81 | 1.53±1.03 |

There are no statistically significant differences in CS on TEOAEs over 1–5 kHz frequencies and average value between acute and chronic tinntius groups for the dominant ear and non-dominant ear (all $p > 0.05$). Analyses were performed by the two-way repeated measures ANOVA.

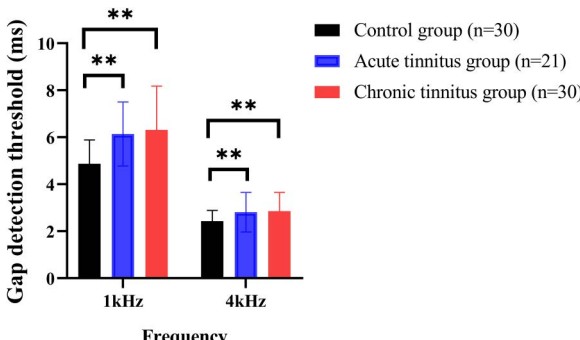

**Fig 2. Comparison of gap detection threshold between the three groups.** There were significant effect of groups and frequency in the gap detection threshold. Analyses were performed by the two-way repeated measures ANOVA. **$p < 0.01$.

For MHINT SRT, a two-way repeated-measures ANOVA (group × ear) showed a significant group effect ($F_{(2,156)} = 35.54$, $p < 0.001$), but no ear effect ($F_{(1,156)} = 0.603$, $p = 0.439$) (Fig 3A). Post hoc tests revealed both tinnitus groups had higher SRTs than controls ($p < 0.001$), with no difference between the acute tinnitus and chronic tinnitus groups ($p = 0.437$). A separate analysis for unilateral tinnitus patients found a significant ear effect ($F_{(1,33)} = 26.720$, $p < 0.001$), but no group effect ($F_{(1,33)} = 1.425$, $p = 0.241$) (Fig 3B).

### 3.3 Correlation of variables in the tinnitus group

Table 3 presents the correlations between gap detection thresholds, SRT in the dominant ear, and tinnitus characteristics in the acute tinnitus and chronic tinnitus groups. Spearman correlation analyses revealed no significant correlations between the average CS on TEOAEs (average left ear and right ear), gap detection threshold, MHINT SRT, and THI and VAS scores, except for a significant correlation between SRT in the dominant ear and THI ($r = 0.669$, $p = 0.009$) in the acute tinnitus group.

In the chronic tinnitus group, there were no significant correlations between the average CS on TEOAEs, gap detection threshold, SRT in the dominant ear, and THI and VAS scores (all $p > 0.05$). We further analyzed and found a significant positive correlation between the emotional scores, catastrophic scores of THI questionnaire and SRT in the dominant ear

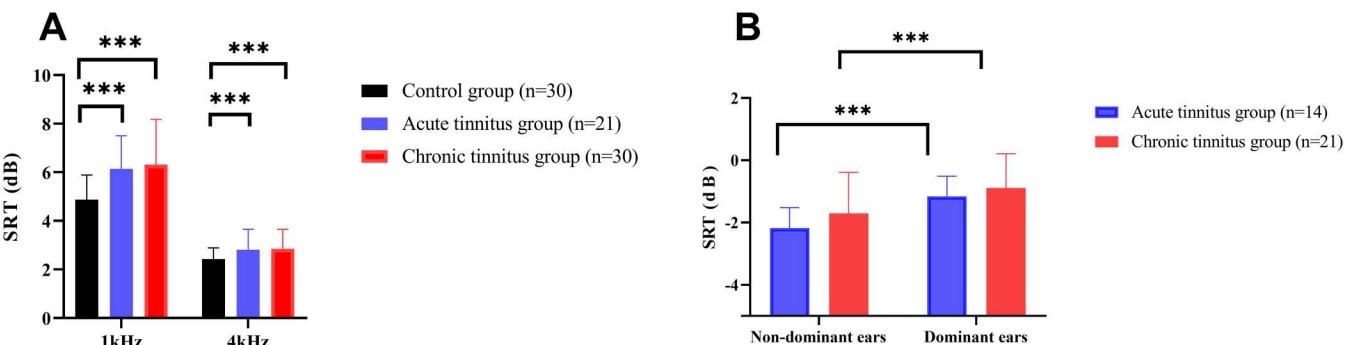

**Fig 3. Comparison of MHINT speech reception thresholds (SRT) across groups and ear dominance.** (A) SRTs differed significantly among the three groups (controls, acute tinnitus, chronic tinnitus) based on one-way ANOVA, with no significant difference between left and right ears. (B) In tinnitus patients, two-way repeated measures ANOVA revealed a significant difference in SRTs between dominant and non-dominant ears, but not between the acute and chronic tinnitus groups. Analyses were conducted using one-way ANOVA (A) and two-way repeated measures ANOVA (B). ***$p < 0.001$.

**Table 3. Spearman correlations among auditory measures and tinnitus scores.**

| | Acute tinnitus group (n = 14) | | Chronic tinnitus group (n = 21) | |
|---|---|---|---|---|
| | THI | VAS | THI | VAS |
| **Average CS on TEOAEs** | 0.197 | −0.419 | 0.023 | 0.006 |
| **GDT threshold at 1kHz** | −0.173 | 0.218 | 0.058 | −0.088 |
| **GDT threshold at 4kHz** | 0.031 | 0.143 | −0.066 | −0.227 |
| **SRT in dominant ear** | **0.669**** | −0.095 | −0.200 | −0.013 |

**p < 0.01.

in the acute tinnitus group ((r = 0.725, p = 0.003; r = 0.678, p = 0.008, respectively), not between functional scores and SRT in the dominant ear (r = 0.427, p = 0.127) (Fig 4).

In addition, we also analyzed the correlation between the duration of tinnitus and THI scores, VAS scores, gap detection threshold, and SRT of MHINT in all tinnitus subjects (Fig 5). Spearman correlation analyses revealed no significant correlations between VAS scores, gap detection threshold, SRT, and duration of Tinnitus (p > 0.05), except for a significant negative correlation between THI scores and duration of Tinnitus (r = −0.476, p < 0.001).

## 4 Discussion

This study is the first to demonstrate behaviorally that central-auditory deficits appear within the first three months of tinnitus and remain unchanged in the chronic stage, despite intact outer-hair-cell function. Adults with normal audiograms but bothersome tinnitus showed markedly reduced contralateral suppression of otoacoustic emissions, prolonged gap detection thresholds, and poorer speech-in-noise perception—deficits of identical magnitude in both acute and chronic tinnitus. High distress in the acute stage is linked to poorer speech recognition performance, linking our data to neuro-imaging evidence that tinnitus engages a multimodal network encompassing auditory cortex and non-auditory hubs such as the amygdala, hippocampus, parahippocampal gyrus, anterior cingulate, precuneus, and dorsolateral prefrontal cortex [16,17,21]. These regions mediate emotion, memory, and attention, helping to explain why heightened early distress magnifies listening difficulties while later limbic adaptation spares further sensory decline.

Emotional distress emerged as a key modulator. In the acute group, higher Tinnitus Handicap Inventory (THI) emotional and catastrophic scores correlated strongly with elevated speech-reception thresholds, suggesting that anxiety-driven attentional capture by the phantom sound diverts cognitive resources from complex listening tasks. Neuro-imaging literature supports this interpretation: heightened coupling between amygdala, hippocampus, and auditory cortex amplifies the salience of tinnitus, while frontal-limbic re-organization over time underpins partial habituation [31–33]. Consistent with this, we observed a negative correlation between tinnitus duration and THI scores; chronic patients reported lower affective load, implying progressive emotional adaptation even though central auditory metrics remained static.

Consistent with previous work, our data indicate that tinnitus leaves peripheral mechanics intact—DPOAE amplitudes were indistinguishable from controls [10,34], —yet produces a robust, early-onset reduction in medial-olivocochlear (MOC) efferent gain. Both acute and chronic groups showed equally diminished contralateral suppression of otoacoustic emissions, implying that the negative-feedback loop is compromised at, or soon after, tinnitus onset and remains stably impaired thereafter. Because MOC suppression is a rapid, largely pre-attentive reflex, its amplitude was unrelated to THI scores, confirming that two partially independent pathological streams coexist: (i) a homeostatic imbalance within subcortical feedback circuitry that weakens efferent inhibition, and (ii) limbic-cortical plasticity driven by affective distress that shapes higher-level auditory processing. Notably, no significant correlation was observed between contralateral inhibition amplitude and either temporal gap detection or speech-in-noise performance. This dissociation likely reflects

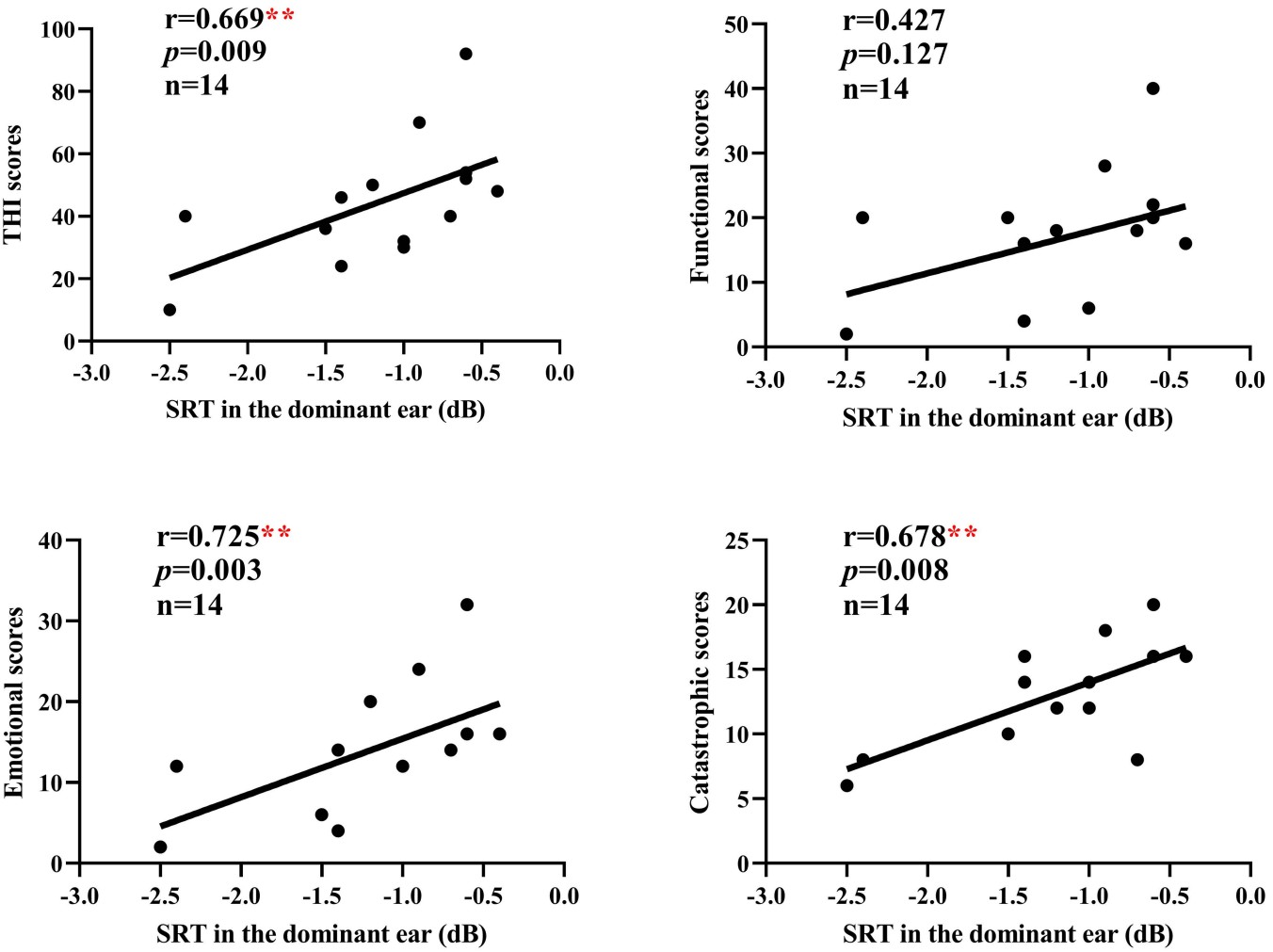

**Fig 4. Correlations between the functional scores, emotional scores, catastrophic scores, THI total scores of THI questionnaire and SRT in the dominant ear in the acute tinnitus group.** Analyses were performed using Spearman's rank correlation. **\*\*p<0.01.**

the hierarchical nature of auditory processing: while MOC suppression reflects peripheral inhibitory control at the level of outer hair cells, temporal discrimination and speech comprehension in noise rely on more complex cortical and cognitive integration. These findings underscore that tinnitus impacts auditory function through both peripheral and central mechanisms, highlighting the need for comprehensive, multi-level assessment and intervention strategies that address both sensory feedback deficits and affective comorbidity.

Although earlier studies—often confounded by age or hearing-loss effects—have yielded mixed results [12,14,35], We observed significantly prolonged gap-detection thresholds at both 1 kHz and the tinnitus frequency, undermining the "filling-in-the-gap" explanation and instead pointing to disrupted top-down temporal processing; elevated, lateralized speech-reception thresholds further corroborated a higher-order deficit. Moreover, our study addressed the open question of whether central auditory dysfunction progresses with tinnitus chronicity or stabilizes after onset. While some studies suggest chronic tinnitus may involve cumulative maladaptive plasticity or sustained emotional burden that exacerbates auditory dysfunction [10,21,23,24,36],with greater neural changes observed in regions involved in emotional regulation, cognitive control, and multisensory integration—such as the hippocampus, amygdala, and frontal cortex—our findings

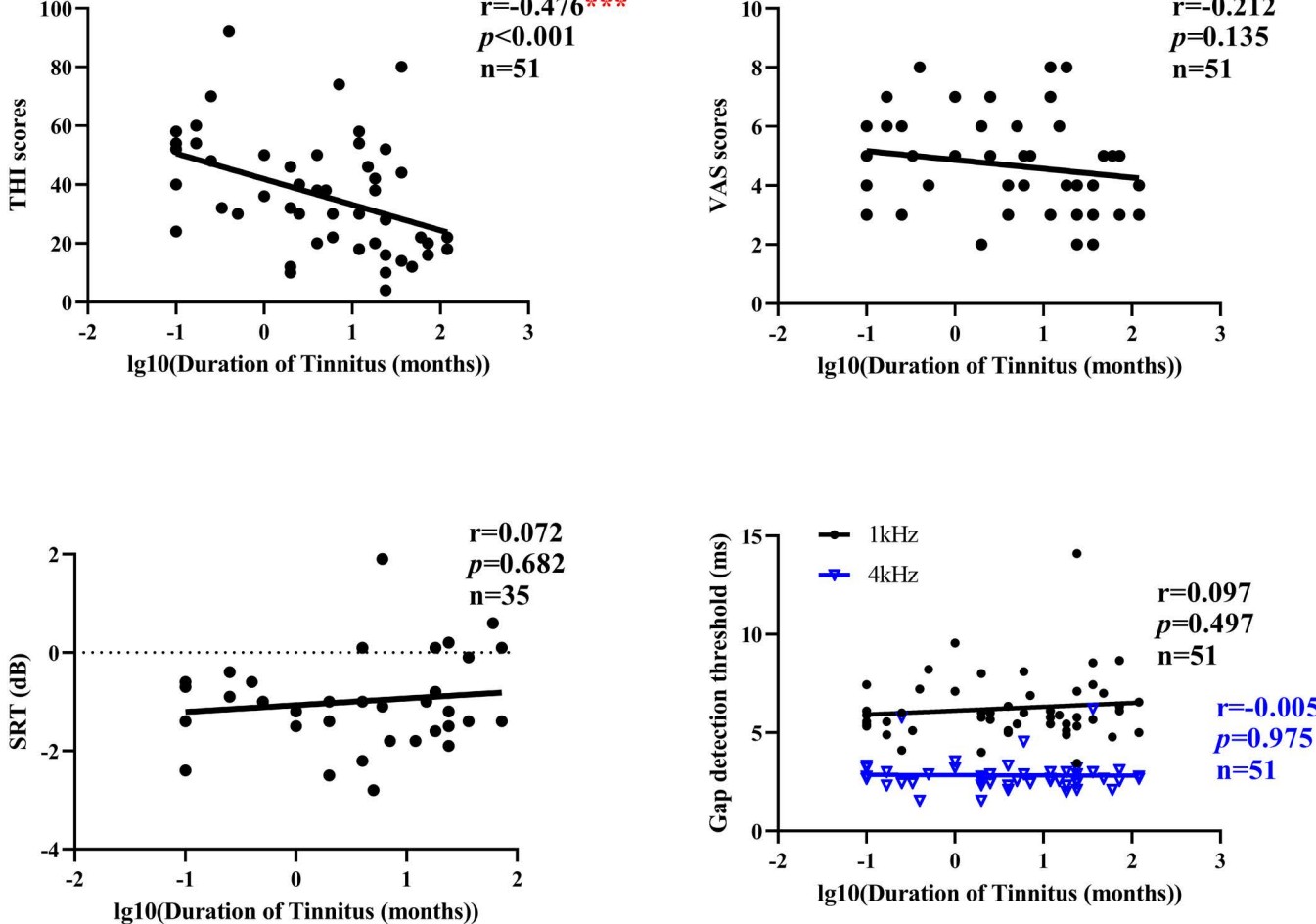

**Fig 5. Correlations between the duration of tinnitus (log10 transformed for visualization purposes) and THI scores, VAS scores, gap detection threshold, and SRT of MHINT.** Analyses were performed using Spearman's rank correlation. ***$p < 0.001$.

support an alternative view: that early central changes emerge shortly after onset but remain relatively stable over time, likely due to partial neural adaptation. Clinically, this finding is encouraging, as it implies that early detection and intervention—particularly targeting emotional distress—may help patients avoid further perceptual deterioration and improve long-term outcomes.

## Limitation

This single-centre study enrolled a modest sample (n = 30 per tinnitus group) of adults aged 18–55 years with normal audiograms; a larger, multi-site cohort spanning younger and older patients—including those with concomitant hearing loss—will be needed to confirm the present findings. Because the design was cross-sectional, we could not follow neuroplastic changes as tinnitus progresses; prospective longitudinal studies coupled with EEG or fMRI are required to chart that trajectory. Emotional distress was assessed only with THI and VAS; adding validated anxiety- and depression scales, and analyzing data with multivariate or machine-learning methods, would provide a fuller account of the psychological factors that interact with auditory function.

## Conclusion

In adults with normal hearing who experience tinnitus, we demonstrate that a stable central-auditory deficit—characterized by weakened efferent suppression, prolonged gap detection, and impaired speech-in-noise recognition—emerges soon after onset and does not deteriorate with chronicity. Its perceptual burden, however, varies with emotional load: high distress in the acute stage is linked to poorer auditory performance, whereas later emotional adaptation lessens this interference. Although causality cannot be inferred from a cross-sectional design, these findings generate the testable hypothesis that effective emotional management might help disrupt the distress-auditory feedback loop and could prevent further functional decline. Recognizing tinnitus as a multisystem condition underscores the need to incorporate psychological support into standard care pathways, aiming to alleviate distress and improve auditory-related quality of life.

## Supporting information

**S1 Table. Demographic information and test results for study participants.** This Excel file includes demographic information and test results for all study participants.
(XLSX)

## Author contributions

**Conceptualization:** Qian Zhou, Wenling Jiang, Zhiwu Huang.

**Funding acquisition:** Zhiwu Huang.

**Investigation:** Qian Zhou, Wenling Jiang, Haibin Sheng, Qinjie Zhang, Dian Jin, Haifeng Li, Meiping Huang, Lu Yang, Yan Ren.

**Methodology:** Qian Zhou, Wenling Jiang, Haibin Sheng, Qinjie Zhang, Dian Jin, Haifeng Li, Meiping Huang, Lu Yang, Yan Ren.

**Project administration:** Zhiwu Huang.

**Supervision:** Zhiwu Huang.

**Writing – original draft:** Qian Zhou, Wenling Jiang.

**Writing – review & editing:** Qian Zhou, Wenling Jiang, Haibin Sheng, Zhiwu Huang.

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
