## [Decision Letter · Decision Letter 0]

PONE-D-25-17398Does Tinnitus and Emotional Distress Influence Central Auditory Processing? A Comparison of Acute and Chronic Tinnitus in Normal-Hearing IndividualsPLOS ONE

Dear Dr. Huang,

Thank you for submitting your manuscript to PLOS ONE. After careful consideration, we feel that it has merit but does not fully meet PLOS ONE’s publication criteria as it currently stands. Therefore, we invite you to submit a revised version of the manuscript that addresses the points raised during the review process.

We look forward to receiving your revised manuscript.

Kind regards,

Hantong Hu

Academic Editor

PLOS ONE

 [This work was supported by the Shanghai Science and Technology Commission [Grant number 22Y11902000]; the Shanghai Jiao Tong University Medical College Affiliated Ninth People's Hospital Horizontal Project [Grant number JY2023-004].].

4. In the online submission form, you indicated that [he data supporting this study’s findings are not available to the public due to participants’ confidentiality but are available from the corresponding authors on request.].

Additional Editor Comments (if provided):

Reviewers' comments:

Reviewer's Responses to Questions

**Comments to the Author**

1. Is the manuscript technically sound, and do the data support the conclusions?

Reviewer #1: Yes

Reviewer #2: Yes

Reviewer #3: Partly

2. Has the statistical analysis been performed appropriately and rigorously? 

Reviewer #1: Yes

Reviewer #2: Yes

Reviewer #3: Yes

3. Have the authors made all data underlying the findings in their manuscript fully available?

Reviewer #1: Yes

Reviewer #2: Yes

Reviewer #3: No

4. Is the manuscript presented in an intelligible fashion and written in standard English?

Reviewer #1: Yes

Reviewer #2: Yes

Reviewer #3: Yes

5. Review Comments to the Author

Reviewer #1: The manuscript is well written with appropriate language. The title is impartial and focused. The abstract is concise and clear. Specific aim is stated and the introduction is clear and covering the background of the topic. The methods and results as well as the statistics are appropriate and the data interpretation is clear and well formulated. The discussion though long but logical and links findings to the existing literature. The significance and limitations are well stated. The references were recent and well organized. The tables and figures are clear and illustrative. Finally, the conclusion was relatively short, clear and well stated.

Overall, this is a relevant manuscript with well-conducted experiments and valuable clinical implications. The findings suggesting that emotional distress, particularly in the acute phase, may exacerbate auditory processing difficulties, highlight the importance of early psychological support. This is an important take-home message for clinicians.

Therefore, this manuscript can be considered appropriate and fit for publication.

Reviewer #2: Dear authors .

Congratulations on your work. You have effectively connected tinnitus, central auditory dysfunction, and emotional distress, which is an important area of research. However, I believe there are opportunities to further strengthen the scientific impact of your study. While the connections made are solid, the work could benefit more scientific exclusivity and the presentation of more disruptive hypotheses. Not your pourpose here but as an idea, why not to integrate new technologies and more precise emotional assessment tools ?

The Materials and Methods section demonstrated technical rigor in auditory testing in a well-controlled environment. However, I believe the study would have been more robust with a larger sample size, longitudinal follow-up, and a more comprehensive emotional assessment as said above ; I also feel that neurofunctional measures, although not your original purpose, could have further enriched the study. This could certainly be an excellent direction for future research.

Acute and chronic tinnitus patients shown significant deficits in central auditory processing, such as elevated gap detection thresholds, poorer speech-in-noise recognition, and reduced contralateral TEOAE suppression, wich validates the central auditory impairment. However, the minimal differences between the acute and chronic phases of tinnitus could have been better explored using advanced neuroimaging methods, such as EEG or functional MRI. As well including the big data approaches to tracking neuroplastic changes over time. Additionally, the correlational analyses would have benefitted from multivariate control to account for other influencing factors.

The correlation between emotional scores on the THI and speech-in-noise performance is important, yet the reliance on THI and VAS alone limits the psychological insight. Incorporating validated scales (in degrees) for anxiety and depression, alongside advanced techniques like machine learning models to predict clinical evolution, could offer a more nuanced emotional analysis.

To conclude , your study brings valuable descriptive evidence regarding the impact of tinnitus on central auditory processing, it remains largely descriptive with very limited innovation. It would benefit go deeper in exploration through neuroimaging and more comprehensive emotional assessments ...

Once again, congratulations on the work, and I look forward to seeing how future studies in this area evolve.

Reviewer #3: The authors study the relationship of tinnitus, emotional distress, and changes in central processing. This is an interesting and important topic, however the study currently has several flaws that I will address in the following:

General comments

The main issue is that the objectives, hypothesis, and study design are not clearly described. The introduction ends with a rather vague statement about the goals of the study, leaving many open questions regarding then study design. Therefore, there should be a dedicated paragraph at the end of the introduction clearly describing why the different measures were used and what the specific research question of this study was. This needs to be more clearly described. Please modify.

The second main issue is that the group definitions are not clear. How was acute and chronic tinnitus differentiated? Why was this distinction made in the first place? How is it defined? How was the presence of tinnitus clinically assessed, and how were the participants recruited? Why was the cut-off of 3 months chosen? Please explain in more detail. It would also be important here to provide an approximate information on how long the chronic tinnitus patients experienced tinnitus.

Another issue is that there was a control group included, however in the presentation of the results and the tables often the results for the control group is not shown. Therefore it is not fully clear why the control group was included, This is not really clear throughout the study. It is not clear what the control group is for, as these were not included as its discussion os often incoherent. Please improve this and explain.

A fourth issue is that the discussion section is quite long and in many parts redundant, with redundant description of the conclusions (twice) and an additional significance statement, which should be condensed and combined into one paragraph or integrated in the general discussion. Otherwise the discussion is confusing for the reader.

A style aspect that too many abbreviations are used which only makes the text hard to read. I do not think it is necessary to abbreviate acute tinnitus (AT) and chronic tinnitus (CT). Also the abbreviation for the control group as CL is not intuitive. Try to reduce the use of abbreviations throughout the text, the reader will thank you for that.

Please include a statement on data availability

Specific comments:

Line 80: It is not clear how the topic of neuroplasticity relates to the previous paragraphs. It might be one of multiple reasons for tinnitus. Here you state that tinnitus is an adaptive response due to reduced peripheral input.This hypothesis needs to be elaborated and explained on in more detail.

Line 87: It is not fully clear why emotions should affect neuroplasticity? Why would emotional distress accelerate the process? What is the biological basis for this? How does it relate to neuroplasticity? This should be explained clearly and supported with adequate references. The hypothesis is not well motivated and the rationale for this is not clear. Please explain.

Line 98: Why does chronic tinnitus lead to significant neural plasticity changes and acute does not? This sentence is not clear. What are the correlates for these neural plasticity changes? What exactly is changed? Please specify.

Line 102: How is the transition from acute to chronic tinnitus defined? What is the exact clinical distinction. Please explain. Please also address the potential different causes for short acute tinnitus and chronic tinnitus.

Line 107: How is the role of emotional distress and neural plasticity quantified here? The exact goal and objective of this study is not fully clear. This needs to be more clearly described. What is the objective and with which methods is this objective going to be addressed. This should be explained more clearly.

Line 114: How were the participants recruited? What background did they have? Why were only young subjects recruited? Please explain in more detail.

Line 129: How was normal hearing assessed? What were the results of the audiometry? Why are these not presented as an summary statistic for the groups? Please include.

Line 172: SL or SPL?

Line 193: dB SL? Do you mean SPL? Were a special Matlab program used to perform the GDT task? Was it custom programmed? Were the stimuli presented via speakers? Or headphones? Was the gap ramped? Please include more information.

Line 214: There are no audiological findings included in Table 1. Please check and explain. Also why were the psychological tests not performed in the control group as a verification? Please explain. Are the p-values reported in table the results of the ANOVA? Then more information about the ANOVA statistics should be reported? What test was used?

Line 218: The TEOAE results seem to be presented in Table 2. This is confusing. Please check which Tables show which information, and provide the correct information and description.

Line 220: Please provide adequate reporting of ANOVA results, including F-statistic and degrees of freedom. Please do so also for the following results where this applies.

Line 220: Why were THI tests and VAS not tested in the control group? How was checked, whether the control group did not have tinnitus?

Line 229: What statistical test was used to compare these results? Please explain and include this information.

Line 231: Here the ANOVA statistic is reported. Why was it not done in the previous cases?

Line 245: In which figure are the results of the GDT test shown? Figure 2? This is not referenced to in the text? The Y-axis label in figure should be changed to Gap detection threshold [ms]. Currently it says ‘time’ which is misleading, as it actually depicts the GDT. Why do the sample sizes differ from the sample sizes described in the methods? 30/30/30 vs. 30/21/30?

Line 291: Why should there a difference in acute tinnitus and chronic tinnitus in the first place. This is not fully clear, and should be explained more clearly.

Line 299: How is it linked to the multimodality network? On what research is this assumption based, please provide some references. Which non-auditory regions does it affect? Please elaborate with adequate references.

Line 410-412: This is an interpretation, but this is not shown in this study. Please tone this down, and point out that this is a hypothesis. This is a bit of an overinterpretation of the results. Although they show a relationship here, they to not show causal effects between the different variables. The assumption that the emotional management directly affects the central auditory system is an speculation.

Line 420: It is not fully clear what is the novel aspect of this study, because it is already known that managing the psychological components of tinnitus is already known. What is the specific new contribution of the current study? This should be pointed out in more detail.

Line 438: The limitations are not discussed in enough detail. How does the small sample size affect the generalisability? Why is the current sample not generalisable, for example due to age. What type of cohorts need to be included? Please elaborate.

Line 444: There seem to exist two conclusion section, as 410-418 also already constitutes a conclusion of the results. This is confusing. Combine the two paragraphs into on conclusion at the end to the paper.

6. PLOS authors have the option to publish the peer review history of their article (what does this mean? ). If published, this will include your full peer review and any attached files.

**Do you want your identity to be public for this peer review?** For information about this choice, including consent withdrawal, please see our Privacy Policy .

Reviewer #1: **Yes: ** Mohamed Badr-El-Dine, M.D, Ph.D.

Reviewer #2: No

Reviewer #3: No

---

## [Author Response · Author response to Decision Letter 1]

3 Jun 2025

Editor Comment 1: “Please ensure that your manuscript meets PLOS ONE's style requirements, including those for file naming. The PLOS ONE style templates can be found at …”

Response: We have revised the manuscript to conform to PLOS ONE’s formatting and style guidelines. This includes proper file naming and formatting of headings, figures, tables, and references as per the PLOS templates. All sections have been reviewed for compliance with PLOS style.

Editor Comment 2: “Please provide additional details regarding participant consent. In the ethics statement in the Methods and online submission, specify (1) whether consent was informed, and (2) what type was obtained (written or verbal, and if verbal, how documented and witnessed). If minors were included, state if parental consent was obtained. If consent was waived, note that.”

Response: Thanks for your insightful feedback. We have clarified the Ethics Statement in the Methods to explicitly state the type of consent and that it was informed. In our study, all participants were adults and provided written informed consent (no minors were included, so parental consent was not applicable). The Ethics Statement now reads:

Original: “This study was approved by the Ethics Committee of the Shanghai Ninth People’s Hospital (Approval number: SH9H-2-22-T379-1), located in Shanghai, China. Written consent was obtained for each participant.”

Revised: “This study was approved by the Ethics Committee of the Shanghai Ninth People’s Hospital (Approval number: SH9H-2-22-T379-1). All participants were adults aged 18 years or older, and written informed consent was obtained from each participant prior to inclusion in the study.”

Editor Comment 3: “Thank you for stating in your Funding Statement: ‘This work was supported by the Shanghai Science and Technology Commission [Grant number 22Y11902000]; … Ninth People's Hospital Horizontal Project [Grant number JY2023-004].’ Please provide an amended statement that declares all funding or sources of support (external or internal) received during this study, as detailed in our guide for authors. Please also include the statement ‘There was no additional external funding received for this study.’ in your updated Funding Statement.”

Response: We have amended the Funding Statement to list all sources of support and to include the required declaration of no additional funding. The revised Funding Statement now explicitly names the funding sources and adds that no other external funding was received. This update will be included in our cover letter and the manuscript. Specifically:

Original Funding Statement: “This work was supported by the Shanghai Science and Technology Commission [Grant number 22Y11902000]; the Shanghai Jiao Tong University Medical College Affiliated Ninth People's Hospital Horizontal Project [Grant number JY2023-004].”

Revised Funding Statement: “This work was supported by the Shanghai Science and Technology Commission (Grant No. 22Y11902000) and the Shanghai Jiao Tong University Medical College Affiliated Ninth People's Hospital Horizontal Project (Grant No. JY2023-004). There was no additional external funding received for this study.”

All sources of support are now disclosed, and the statement of no other funding has been added as requested.

Editor Comment 4: “In the online submission form, you indicated that ‘the data supporting this study’s findings are not available to the public due to participants’ confidentiality but are available from the corresponding authors on request.’ PLOS data policy requires all data underlying the findings to be fully available without restriction (with rare exception). If data cannot be made public for ethical or legal reasons, please explain and request an exemption.”

Response: We have revised the Data Availability Statement to comply with PLOS ONE’s data policy. All data underlying our findings have now been made freely available in anonymized form. Specifically, we have prepared a de-identified dataset as Supporting Information and indicated this in the manuscript. The Data Availability section of our revised manuscript now states that “All relevant data are within the manuscript and its Supporting Information files.” If necessary, we will also deposit the data in a public repository and provide the DOI. We have removed the previous restriction (“available on request”) by anonymizing sensitive information, thereby addressing the confidentiality concern. In summary, all data are now publicly available either in the supplementary files or via a public repository, in accordance with PLOS policy.

Editor Comment 5: “PLOS requires an ORCID iD for the corresponding author. Please ensure you have an ORCID iD and that it is validated in Editorial Manager (by clicking the Fetch/Validate link next to the ORCID field).”

Response: We have verified that the corresponding author has an ORCID iD and it is validated in the Editorial Manager system. The ORCID iD has been linked and authenticated as required.

Editor Comment 6: “Please include captions for your Supporting Information files at the end of your manuscript, and update any in-text citations to match. See our Supporting Information guidelines for more information.”

Response: We have added a section at the end of the manuscript listing captions for all Supporting Information files. Each supporting figure or table (e.g., any supplementary audiograms or data files) now has a caption (labeled as S1, S2, etc.) included after the References section. We also cross-checked and updated any in-text references to these supplementary items to ensure they are correctly labeled (e.g., “S1 Table”, “S1 Figure”). This addition ensures compliance with PLOS ONE’s guidelines on Supporting Information. we have created a minimal data supplement to fulfill data sharing requirements as noted above.

Reviewer #1:

Comment (Reviewer #1): “The manuscript is well written with appropriate language. The title is focused, the abstract concise and clear. The introduction covers the background. Methods, results, and statistics are appropriate and clearly interpreted. The discussion, though long, is logical and links findings to existing literature. Significance and limitations are well stated. References are recent and well organized. Tables and figures are clear. The conclusion is short, clear, and well stated. Overall, this is a relevant manuscript with well-conducted experiments and valuable clinical implications. The findings (emotional distress in acute phase exacerbating difficulties) highlight the importance of early psychological support – an important take-home message for clinicians. Therefore, this manuscript can be considered appropriate and fit for publication.”

Response: We sincerely thank Reviewer #1 for the positive and encouraging assessment of our work. We are pleased that the reviewer found the manuscript well-written, clear, and clinically relevant. We have taken note of the reviewer’s remarks regarding the length of the Discussion and the clarity of the Conclusions. In line with suggestions from the other reviewers, we streamlined the Discussion to remove redundancies and combined the concluding remarks into a single Conclusion section to improve clarity (see our responses to Reviewer #3 for details). We also performed minor edits for clarity and consistency throughout the manuscript. These changes further improve readability while preserving the content that Reviewer #1 praised. We appreciate Reviewer #1’s supportive comments and have maintained the strengths noted (focused title, clear abstract, thorough introduction, etc.) in the revised manuscript. Once again, thank you for your positive feedback and for highlighting the key message about early psychological support, which we have ensured remains prominent in the revised Conclusion.

Reviewer #2:

Comment (Reviewer #2): *“Dear authors: Congratulations on your work. You have effectively connected tinnitus, central auditory dysfunction, and emotional distress, which is an important area of research. However, I believe there are opportunities to further strengthen the scientific impact of your study. While the connections made are solid, the work could benefit from more scientific exclusivity and disruptive hypotheses. (Not your purpose here but as an idea, why not integrate new technologies and more precise emotional assessment tools?)

The Materials and Methods section demonstrated technical rigor in auditory testing in a well-controlled environment. However, the study would have been more robust with a larger sample size, longitudinal follow-up, and a more comprehensive emotional assessment as said above. I also feel that neurofunctional measures, although not your original purpose, could have further enriched the study – this could be an excellent direction for future research.

Acute and chronic tinnitus patients showed significant deficits in central auditory processing (elevated GDT thresholds, poorer speech-in-noise, reduced TEOAE suppression), which validates central auditory impairment. However, the minimal differences between acute and chronic phases could have been better explored using advanced neuroimaging methods (EEG or fMRI), as well as big data approaches to track neuroplastic changes over time. Additionally, the correlational analyses would have benefitted from multivariate control to account for other influencing factors.

The correlation between emotional scores (THI and VAS) and speech-in-noise performance is important, yet reliance on THI and VAS alone limits psychological insight. Incorporating validated scales for anxiety and depression, alongside advanced techniques like machine learning models to predict clinical evolution, could offer a more nuanced emotional analysis.

To conclude, your study provides valuable descriptive evidence regarding the impact of tinnitus on central auditory processing, but it remains largely descriptive with very limited innovation. It would benefit from deeper exploration through neuroimaging and more comprehensive emotional assessments... Once again, congratulations on the work, and I look forward to seeing how future studies in this area evolve.”*

Response: We appreciate your positive appraisal of our methodological rigour and your thoughtful suggestions for strengthening the scientific reach of our work. Although the present study was designed as a behavioral comparison, we fully agree that advanced neuro-imaging, richer emotion scales, and multivariate or machine-learning analyses would deepen future investigations. The following changes on Limitation have been made:

Revised

“Limitations. This single-centre study enrolled a modest sample (n = 30 per tinnitus group) of adults aged 18–55 years with normal audiograms; a larger, multi-site cohort spanning younger and older patients—including those with concomitant hearing loss—will be needed to confirm the present findings. Because the design was cross-sectional, we could not follow neuroplastic changes as tinnitus progresses; prospective longitudinal studies coupled with EEG or fMRI are required to chart that trajectory. Emotional distress was assessed only with THI and VAS; adding validated anxiety- and depression scales, and analysing data with multivariate or machine-learning methods, would provide a fuller account of the psychological factors that interact with auditory function.”

These additions directly reflect your recommendations while keeping the manuscript focused and concise. Thank you again for your constructive guidance.

Reviewer #3:

General Comment 1 (Objectives and Study Design): “The main issue is that the objectives, hypothesis, and study design are not clearly described. The introduction ends with a rather vague statement about the goals of the study, leaving many open questions regarding the study design. There should be a dedicated paragraph at the end of the introduction clearly describing why the different measures were used and what the specific research question of this study was. This needs to be more clearly described. Please modify.”

Response: We agree with the reviewer that the introduction should more clearly articulate the specific objectives and rationale of our study, including why we chose measures. We have added a dedicated concluding paragraph to the Introduction to explicitly state our research questions and the study design. In the revised Introduction, we outline the purpose of the study, the key auditory measures used, and the hypotheses regarding acute vs. chronic tinnitus. Specifically, we now explain that we used the gap detection test (GDT), the Mandarin Hearing in Noise Test (MHINT), and contralateral suppression of TEOAEs to assess different aspects of central auditory processing (temporal resolution, speech-in-noise recognition, and efferent function, respectively) in acute and chronic tinnitus patients versus controls. We also clearly state the hypothesis that emotional distress may influence these auditory processing outcomes. This added paragraph (last paragraph of Introduction) provides a focused summary of the study’s aims and design, as requested.

Original (Lines 104–112): “To our knowledge, this is the first study to investigate differences in central auditory processing function between patients with acute and chronic tinnitus. The purpose of this study is to evaluate whether tinnitus affects central auditory function in individuals with normal hearing and to explore the role of emotional distress and neural plasticity in this process. If functional changes are present, this study seeks to determine whether these changes are stable or influenced by dynamic processes over time. Understanding these mechanisms will provide valuable insights into tinnitus and offer important clinical implications for diagnosing and managing tinnitus in individuals with normal hearing.”

Revised: “To our knowledge, this is the first study to investigate differences in central auditory processing function between patients with acute and chronic tinnitus. We employed the gap detection test (GDT), Mandarin Hearing in Noise Test (MHINT), and contralateral suppression of TEOAEs to assess temporal resolution, speech-in-noise recognition, and efferent auditory function, respectively. The purpose of this study is to evaluate whether tinnitus affects central auditory function in normal-hearing individuals and to explore how emotional distress and neural plasticity may contribute to any observed changes. If functional changes are present, we further seek to determine whether these changes differ between acute and chronic stages of tinnitus or remain stable over time. Understanding these mechanisms will provide valuable insights into how tinnitus evolves and offer important clinical implications for managing tinnitus.”

This revised Introduction now clearly identifies what we are testing and why, directly addressing the reviewer’s concern.

General Comment 2 (Group Definitions – Acute vs. Chronic Tinnitus): “The second main issue is that the group definitions are not clear. How was acute and chronic tinnitus differentiated? Why was this distinction made in the first place? How is it defined? How was the presence of tinnitus clinically assessed, and how were the participants recruited? Why was the cut-off of 3 months chosen? Please explain in more detail. It would also be important here to provide approximate information on how long the chronic tinnitus patients experienced tinnitus.”

Response: We appreciate the opportunity to clarify our group definitions and recruitment methods. We now define acute tinnitus as < 3 months and chronic tinnitus as ≥ 3 months in Methods. The 3-month cut-off follows clinical practice (Japanese Guideline 2020; German Guideline 2022) and recent research (Zhang et al., 2023). We have added this explanation to the manuscript to justify the distinction.

We also now describe how tinnitus presence was assessed and how participants were recruited. All tinnitus patients were initially evaluated by an otolaryngologist and underwent audiological examination (to confirm normal hearing) as well as tinnitus pitch and loudness matching. We recruited tinnitus participants from the Otolaryngology clinic of our hospital (Shanghai Ninth People’s Hos

---

## [Decision Letter · Decision Letter 1]

Does Tinnitus and Emotional Distress Influence Central Auditory Processing? A Comparison of Acute and Chronic Tinnitus in Normal-Hearing Individuals

PONE-D-25-17398R1

Dear Dr. Huang,

We’re pleased to inform you that your manuscript has been judged scientifically suitable for publication and will be formally accepted for publication once it meets all outstanding technical requirements.

Kind regards,

Hantong Hu

Academic Editor

PLOS ONE

Additional Editor Comments (optional):

Reviewers' comments:

Reviewer's Responses to Questions

**Comments to the Author**

1. If the authors have adequately addressed your comments raised in a previous round of review and you feel that this manuscript is now acceptable for publication, you may indicate that here to bypass the “Comments to the Author” section, enter your conflict of interest statement in the “Confidential to Editor” section, and submit your "Accept" recommendation.

Reviewer #1: All comments have been addressed

Reviewer #3: All comments have been addressed

2. Is the manuscript technically sound, and do the data support the conclusions?

Reviewer #1: Yes

Reviewer #3: Partly

3. Has the statistical analysis been performed appropriately and rigorously? 

Reviewer #1: Yes

Reviewer #3: Yes

4. Have the authors made all data underlying the findings in their manuscript fully available?

Reviewer #1: Yes

Reviewer #3: Yes

5. Is the manuscript presented in an intelligible fashion and written in standard English?

Reviewer #1: Yes

Reviewer #3: Yes

6. Review Comments to the Author

Reviewer #1: After thoroughly reviewing the authors’ responses to the different reviewers’ comments, I find that all concerns and suggestions have been adequately addressed. The authors have made substantial improvements by reformatting, rephrasing, and clarifying all points as requested.

Upon re-evaluating the revised manuscript, including the Introduction, Materials and Methods, Results, Discussion, and Conclusion, I believe that the manuscript now meets the publication standards and formatting requirements of PLOS ONE.

Therefore, I consider this revised manuscript to be suitable for publication.

Reviewer #3: I thank the authors to address all my comments and including appropriate changes and modifications in the manuscript.

7. PLOS authors have the option to publish the peer review history of their article (what does this mean? ). If published, this will include your full peer review and any attached files.

**Do you want your identity to be public for this peer review?** For information about this choice, including consent withdrawal, please see our Privacy Policy .

Reviewer #1: **Yes: ** Mohamed M.K. badr-El-Dine, M.D., Ph.D.

Reviewer #3: No

---

## [Editor Report · Acceptance letter]

PONE-D-25-17398R1

PLOS ONE

Dear Dr. Huang,

I'm pleased to inform you that your manuscript has been deemed suitable for publication in PLOS ONE. Congratulations! Your manuscript is now being handed over to our production team.

Kind regards,

on behalf of

Dr. Hantong Hu

Academic Editor

PLOS ONE